# Industrial brewing yeast engineered for the production of primary flavor determinants in hopped beer

Charles M. Denby[1,2], Rachel A. Li[2,3,4], Van T. Vu[5], Zak Costello[2,4,6], Weiyin Lin[1,2], Leanne Jade G. Chan[2,4], Joseph Williams[7], Bryan Donaldson[8], Charles W. Bamforth [7], Christopher J. Petzold[2,4], Henrik V. Scheller[2,3,9], Hector Garcia Martin [2,4,6] & Jay D. Keasling [1,2,4,5,10,11]

Flowers of the hop plant provide both bitterness and "hoppy" flavor to beer. Hops are, however, both a water and energy intensive crop and vary considerably in essential oil content, making it challenging to achieve a consistent hoppy taste in beer. Here, we report that brewer's yeast can be engineered to biosynthesize aromatic monoterpene molecules that impart hoppy flavor to beer by incorporating recombinant DNA derived from yeast, mint, and basil. Whereas metabolic engineering of biosynthetic pathways is commonly enlisted to maximize product titers, tuning expression of pathway enzymes to affect target production levels of multiple commercially important metabolites without major collateral metabolic changes represents a unique challenge. By applying state-of-the-art engineering techniques and a framework to guide iterative improvement, strains are generated with target performance characteristics. Beers produced using these strains are perceived as hoppier than traditionally hopped beers by a sensory panel in a double-blind tasting.

[1] California Institute of Quantitative Biosciences (QB3), University of California, Berkeley, CA 94720, USA. [2] Joint BioEnergy Institute, Emeryville, CA 94608, USA. [3] Department of Plant and Microbial Biology, University of California, Berkeley, CA 94720, USA. [4] Lawrence Berkeley National Laboratory, Biological Systems and Engineering Division, Berkeley, CA 94720, USA. [5] Department of Bioengineering, University of California, Berkeley, CA 94720, USA. [6] DOE Agile BioFoundry, Emeryville, CA 94608, USA. [7] Department of Food Science and Technology, University of California Davis, Davis, CA 95616, USA. [8] Lagunitas Brewing Company, Petaluma, CA 94954, USA. [9] Lawrence Berkeley National Laboratory, Environmental Genomics and Systems Biology Division, Berkeley, CA 94720, USA. [10] Department of Chemical and Biomolecular Engineering, University of California, Berkeley, CA 94720, USA. [11] Novo Nordisk Foundation Center for Sustainability, Technical University of Denmark, 2900 Hellerup, Denmark. These authors contributed equally: Charles M. Denby and Rachel A. Li. Correspondence and requests for materials should be addressed to C.M.D. (email: charles@bbsbeer.com) or to J.D.K. (email: jdkeasling@lbl.gov)

During the brewing process, *Saccharomyces cerevisiae* converts the fermentable sugars from grains into ethanol and a host of other flavor-determining by-products. Flowers of the hop plant, *Humulus lupulus* L., are typically added during the wort boil to impart bitter flavor and immediately before or during the fermentation to impart "hoppy" flavor and fragrance (Fig. 1a). Over the past two decades, consumers have displayed an increasing preference for beers that contain hoppy flavor. Hops are an expensive ingredient for breweries to source (total domestic sales have tripled over the past 10 years due to heightened demand) and a crop that requires a large amount of natural resources: ~100 billion L of water is required for annual irrigation of domestic hops and considerable infrastructure is required to deliver water from its source to the farm[1,2]. Further, hops vary considerable in essential oil content, making it challenging to achieve a consistent hoppy taste in beer.

Hop flowers are densely covered by glandular trichomes, specialized structures that secrete secondary metabolites into epidermal outgrowths[3]. These secretions accumulate as essential oil, which is rich in various terpenes, the class of metabolites that impart hoppy flavor to beer. Considerable research has investigated which of these molecules are primarily responsible for this flavor[4]; these studies are complicated by genetic, environmental, and process-level variation[5] and have suggested that the bouquet of flavor molecules contributed to beer by hops is complex. Nonetheless, the two monoterpene molecules linalool and geraniol have been identified as primary flavor determinants by several sensory analyses of hop extract aroma[6-8] and finished beer taste and aroma[7,9-11], and together, they are major drivers of the floral aroma of Cascade hops[9], the most widely used hop in American craft brewing[12]. Previous metabolic engineering efforts have achieved microbial monoterpene biosynthesis in various microbial hosts. Work in a domesticated wine yeast has demonstrated the feasibility of producing monoterpene compounds by biosynthesis in yeast by overexpression of a geraniol synthase from a high-copy plasmid propagated in selective media[13]. However, engineering genetically stable, controlled, precise production of a combination of specific flavorants in any industrial food-processing agent has remained a formidable challenge.

In this work, we create drop-in brewer's yeast strains capable of biosynthesizing monoterpenes that give rise to hoppy flavor in finished beer, without the addition of flavor hops. To achieve this end, we identify genes suitable for monoterpene biosynthesis in yeast; we develop methods to overcome the difficulties associated with stable integration of large constructs in industrial strains; we adapt genetic tools to generate a collection of engineered industrial yeast strains on an unprecedented scale; we develop computational methods to affect precise biosynthetic control and leverage them to create a iterative framework towards target production levels. Ultimately, sensory analysis performed with beer brewed in pilot industrial fermentations demonstrates that engineered strains confer hoppy flavor to finished beer.

## Results

**Identification of yeast-active linalool and geraniol synthases.** The monoterpene synthases that catalyze the reaction of geranyl pyrophosphate (GPP) to linalool and geraniol in hops have not yet been identified[14]. However, genes from other plant species have been shown to encode these activities. To identify a linalool synthase (LIS) gene for functional heterologous expression in yeast, we expressed six different plant-derived LIS genes in a lab yeast strain engineered for high GPP precursor supply (Fig. 1b). However, none of the full-length proteins exhibited sufficient activity to achieve target monoterpene concentrations in finished beer (Fig. 1c). In plants, monoterpenes are biosynthesized in chloroplasts; plant monoterpene synthases, therefore, typically contain an N-terminal plastid targeting sequence (PTS) composed of 20–80 polar amino acids, which is cleaved to produce a mature protein. Truncation of the PTS sequence can improve expression and activity of microbially expressed monoterpene synthases[15,16]. However, methods for predicting the optimal PTS truncation site, as well as for predicting portability of enzymes from plant species to yeast are imperfect. We therefore screened candidate LIS variants from different sources and with different truncation sites for increased activity (Fig. 1b and Supplementary Fig. 1). We tested bioinformatically predicted[17] PTS sites and observed a substantial increase in activity for the *Lycopersicon esculentum* LIS (Fig. 1c). We also used a heuristic, structure-based

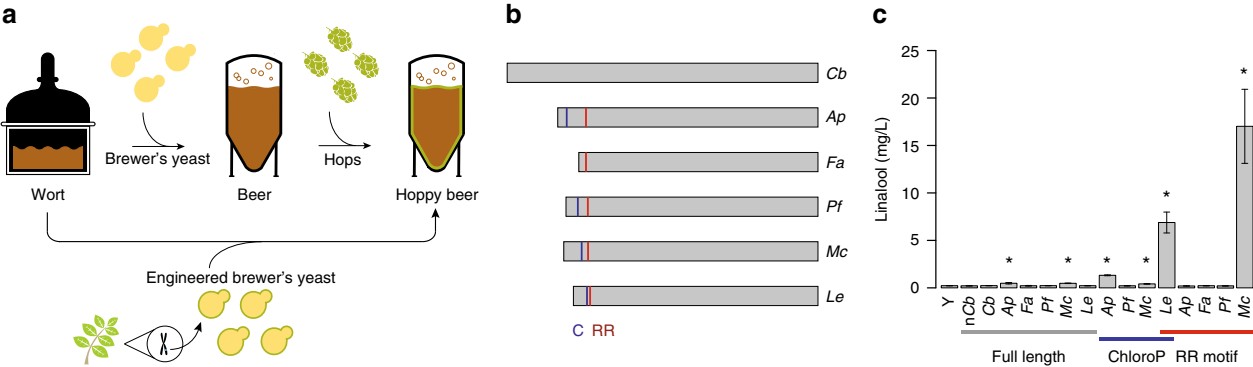

**Fig. 1** Engineering brewer's yeast to express monoterpene biosynthetic pathways thereby replacing flavor hop addition. **a** During the brewing process, *S. cerevisiae* converts wort—a barley extract solution rich in fermentable sugars—into ethanol and other by-products. Hops are added immediately before, during, or after fermentation to impart "hoppy" flavor. Engineered strains produce linalool and geraniol, primary flavor components of hoppy beer, thereby replacing hop additions. **b** Six full-length plant-derived linalool synthase genes, as well as PTS-truncated variants, were expressed on high-copy plasmids. Full-length genes and PTS-truncated genes predicted by either ChloroP (C) or the RR-heuristic method (RR) are indicated by colored lines. **c** Error bars correspond to mean ± standard deviation of three biological replicates. Asterisks indicate statistically significant increases in monoterpene production compared with the control strain (Y) as determined by a *t*-test using *p*-value <0.025. The LIS from the California wildflower *Clarkia breweri* has been shown to increase production of linalool when heterologously expressed in plants[47] and in yeast[48]. However, when *C. breweri* LIS was expressed, either with native codons (nCb) or "yeast-optimized" codons (Cb), linalool was not detected. The *Mentha citrata* LIS (Mc) truncated at the RR motif was identified as sufficiently active to allow for monoterpene production at levels characteristic of commercial beer and was chosen for integration into brewer's yeast strains

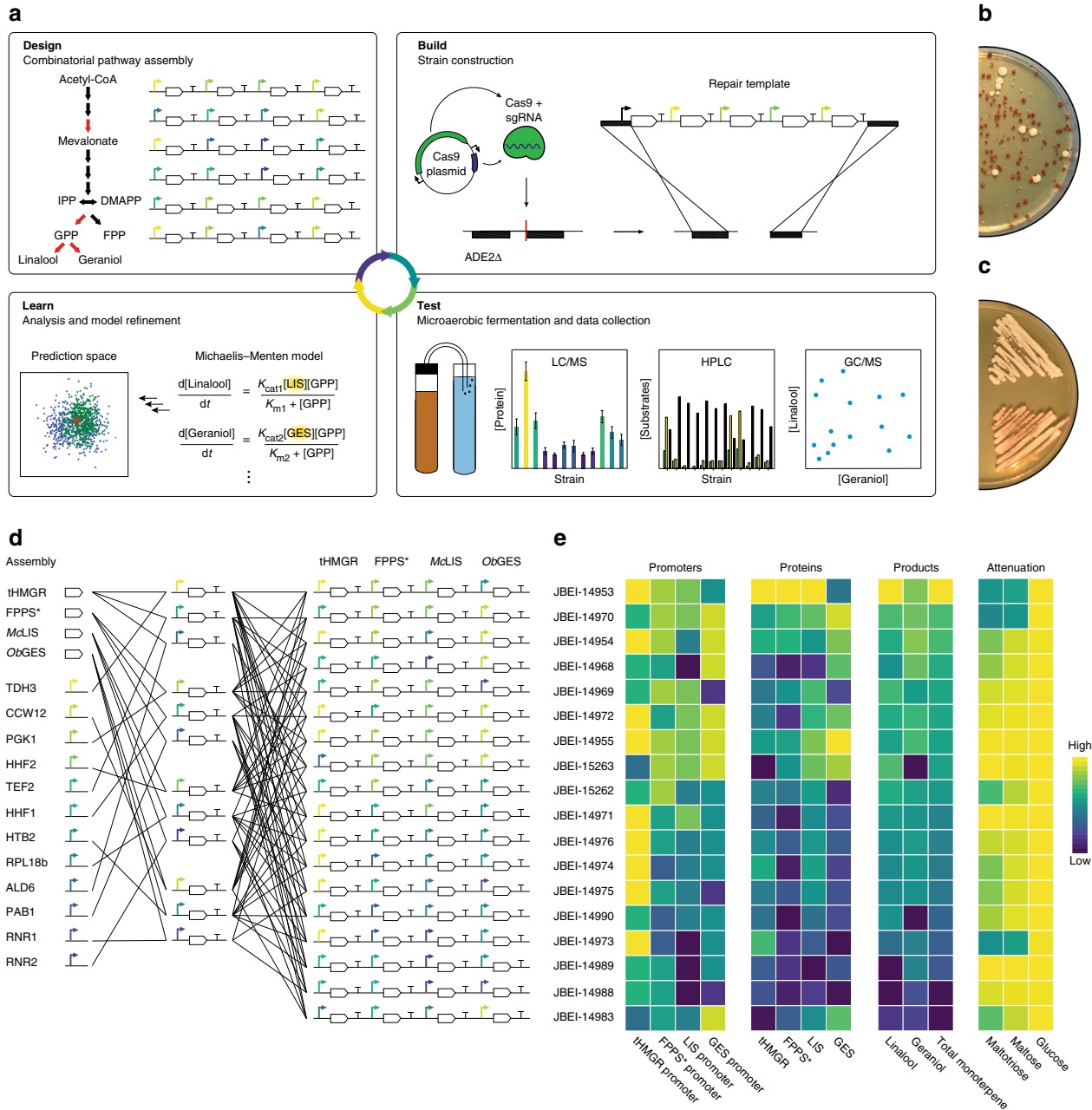

**Fig. 2** Iterative improvement of strain design towards targeted monoterpene production levels. **a** Schematic of design–build–test–learn cycle. Design: Constructs were designed by combining yeast toolkit parts (i.e., promoters, terminators, linkers, etc.) with monoterpene biosynthesis pathway genes. Build: Methodology for integrating constructs into brewer's yeast. Note that, for simplicity, only a single allelic copy of the ADE2 locus is diagrammed. The ADE2Δ strain was co-transformed with a Cas9/sgRNA plasmid and repair template, which targeted a double-stranded break (DSB) in the ADE2 3′ sequence. Test: Data were collected using LC/MS, HPLC, and GC/MS. Learn: Correlation analyses informed design principles. Mathematical models were used to evaluate the extent to which design principles improved strain search efficiency. Variables corresponding to measured protein levels are highlighted. **b**, **c** Transformation plate illustrating colorimetric screening method. ADE2 encodes an enzymatic step in purine biosynthesis and its deletion results in the accumulation of a metabolite with red pigment when grown on media containing intermediate adenine concentration. Because the repair template contains the *ADE2* gene, templated DSB repair results in a white colony phenotype. Because brewer's yeasts have multiple allelic copies of ADE2, stable integration requires repair at multiple ADE2Δ genomic loci. White colonies streaked from transformation plates result in either white colony color (**b**) or variegated colony color (**c**); white colony color corresponds to homozygous integration; variegated colony color corresponds to heterozygous integration, illustrating genetic instability of heterozygous allele containing a large DNA construct. **d** Illustration of assembly steps from parts (promoters/genes) to gene cassettes, to repair templates for first iteration strains. Assemblies are simplified for clarity—for detailed description see Supplementary Fig. 3. **e** Relative promoter strengths with corresponding protein and product abundances and sugar consumption (attenuation). Strains are sorted by total monoterpene production

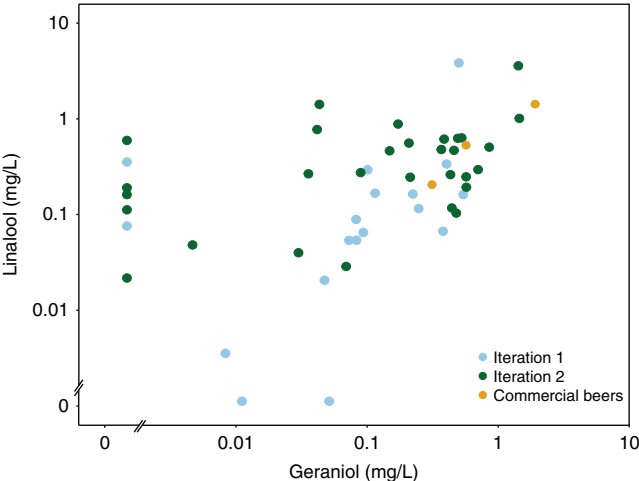

**Fig. 3** Production of monoterpenes by engineered strains. Linalool and geraniol production of engineered yeast strains compared to concentrations found in commercial beers, plotted in $\log_{10}$ space. For relationships between flavor determinant concentration and taste intensity, the logarithm of a stimulus is typically proportional to the logarithm of the perceived intensity, such that the distance between points in $\log_{10}$ space is expected to be directly proportional to the magnitude of taste difference. First and second iteration points represent the mean of three biological replicates. Standard deviation values are listed in Supplementary Table 14. In ascending order of monoterpene concentration, commercial beers are Pale Ale, Torpedo Extra IPA, and Hop Hunter IPA, obtained from the Sierra Nevada Brewing Company

approach for PTS prediction: a conserved double arginine (RR) motif that functions as part of an active-site lid preventing water access to the carbocationic reaction intermediate[18] lies immediately C-terminal to the PTS in several well-characterized terpene synthases[15,16]. We observed the highest linalool titer from the truncated *M. citrata* LIS (t67-*Mc*LIS, Fig. 1c). An analogous screen of six geraniol synthases revealed that the heterologous expression of the full-length protein from *O. basilicum* (*Ob*GES) leads to geraniol production in yeast (Supplementary Fig. 1d).

**Strategy for engineering monoterpene biosynthesis in brewing yeast.** Once we identified monoterpene synthases that were sufficiently active in *S. cerevisiae*, we set out to engineer brewer's yeast strains capable of producing monoterpenes during beer fermentation. Yeast naturally produces the sesquiterpene precursor farnesyl pyrophosphate (FPP) through the ergosterol biosynthesis pathway, as FPP serves as a precursor for essential metabolites including hemes and sterols. While the flux through this pathway is tightly regulated, extensive metabolic engineering efforts have informed key genetic modifications that obviate regulatory checkpoints[19,20] and increase monoterpene precursor supply[21] (Supplementary Fig. 2). HMG-CoA reductase (HMGR) is one of the key rate-limiting steps of the pathway and is controlled by an inhibitory regulatory domain that responds to product accumulation[19]. Overexpression of a truncated form of yeast HMGR lacking the regulatory domain (tHMGR) results in increased flux towards end products[22]. A downstream enzyme, FPP synthase (FPPS), catalyzes the sequential condensation of two isopentyl pyrophosphate molecules with dimethylallyl pyrophosphate. GPP, the immediate precursor of monoterpene biosynthesis, is formed as an intermediate of the sequential reactions. While the high processivity of the wild-type FPPS limits the intracellular availability of GPP, a mutant (FPPS*) has been identified that reduces processivity, thereby increasing

production of GPP-derived end products[21]. Based on these observations, we hypothesized that modulating the expression of tHMGR, FPPS*, t67-*Mc*LIS, and *Ob*GES would result in brewer's yeast strains capable of producing linalool and geraniol during fermentation at concentrations encompassing those typical of finished beer (~0.2 mg/L)[9,23].

In devising a strategy to modulate pathway activity, two challenges were considered. First, de novo design and generation of a collection of multi-gene constructs is difficult, time consuming, and expensive. To circumvent this challenge, we combined an existing toolkit of yeast genetic parts with a Golden Gate assembly strategy for facile design and rapid pathway construction[24] (Fig. 2a, d and Supplementary Fig. 3). Second, incorporating large (i.e., >10 kb) genetically stable DNA constructs into brewer's yeast has not been reported, and is complicated by their ploidy as well as concerns regarding the incorporation of selection markers in food-processing agents. We therefore developed a Cas9-mediated methodology for stable and marker-less pathway integration (Fig. 2a–c, Supplementary Fig. 4, and Supplementary Note 1). Our method leverages a colorimetric assay to screen for positive transformants and allows for macroscopic visualization of successful integration events. Interestingly, this method also allowed us to visualize the high degree of genetic instability associated with heterozygous integration (Fig. 2b, c). By combining the assembly and integration strategies, we were able to generate strains with a diverse set of genetic designs, where each strain contained a unique combination of promoters driving expression of the four modulated genes (Fig. 2d).

**Iterative design refines target monoterpene levels.** Without empirical data, it is difficult to predict the relationship between specific genetic designs and metabolic end-product concentrations[25,26]. To improve search efficiency towards desired monoterpene concentrations, we separated our design–build-test process (Fig. 2a) into two stages, thereby affording us an opportunity to first sample a small subset of design space and then hone subsequent designs towards desired production profiles. An initial set of 18 strains containing promoters predicted to span a wide range of expression strengths were constructed and grown under microaerobic fermentation conditions that mimicked an industrial brewing process (Supplementary Fig. 5). We found that these first iteration strains produced monoterpenes within the range of commercial concentrations, although were generally lower (Fig. 3). Some strains exhibited a reduced fermentation capacity (Supplementary Figs. 6 and 7) including the strains closest to commercial concentrations. However, reduced fermentation capacity did not correlate with monoterpene production, suggesting that the fermentation defects were not primarily due to monoterpene toxicity (Supplementary Note 2).

To further explore the relationship between genetic design and monoterpene production, the relative abundance of the four modulated proteins was measured for each strain during the active phase of fermentation. Protein abundance was strongly correlated with previously characterized promoter strength (Supplementary Table 1 and Supplementary Fig. 8), demonstrating that the qualitative relationship between promoter strengths generally extends from a lab strain grown aerobically in rich medium to a brewing strain grown in industrial brewing conditions. Furthermore, total monoterpene production was correlated with tHMGR and FPPS* abundance and linalool production was correlated with t67-*Mc*LIS abundance, verifying that the selected genes indeed control monoterpene production as anticipated (Fig. 2e and Supplementary Table 1). An interesting

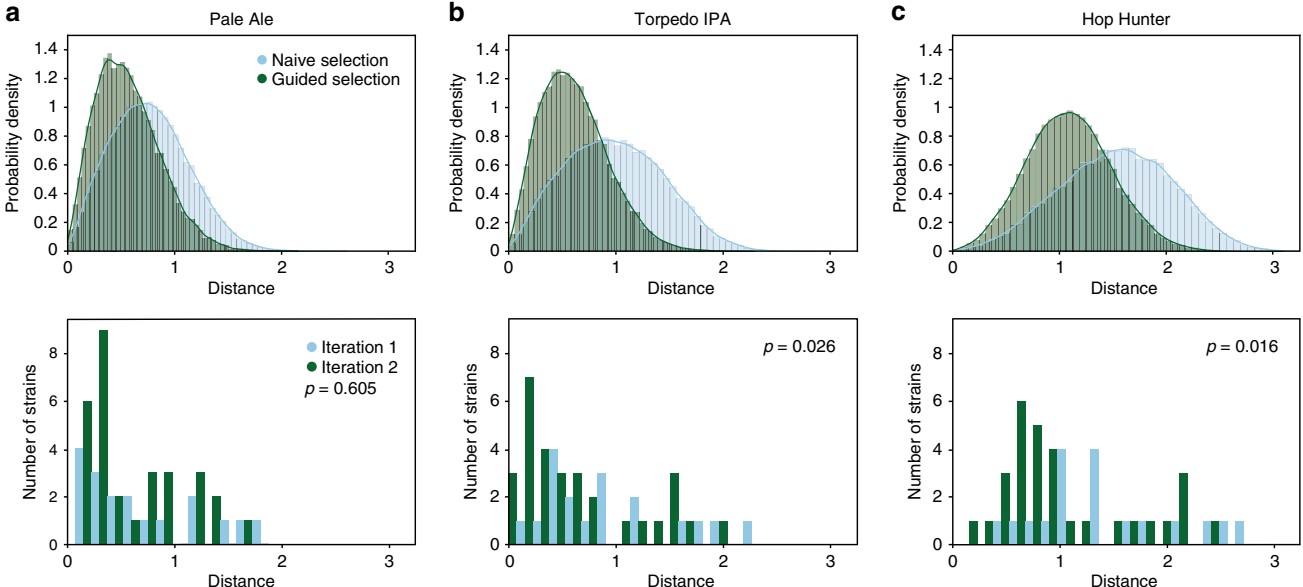

**Fig. 4** Evaluation of iterative genetic design refinement. Simulated (top panel) and measured (bottom panel) distance between engineered strain performance and target performance of commercial beers obtained from Sierra Nevada Brewing Company for first and second iteration strains. Target performance is defined based on Pale Ale (**a**), Torpedo IPA (**b**), and Hop Hunter IPA (**c**). In all cases, second iteration strains are closer to target performance than first iteration strains. *p*-values (*t*-test) reflect the degree of statistical significance with which second iteration strains are closer to target performance then first iteration strains

anomaly was that *Ob*GES abundance was not correlated with geraniol production. We reasoned that since *Ob*GES and t67-*Mc*LIS compete for GPP supply, variation in t67-*Mc*LIS abundance may obscure the relationship between *Ob*GES and geraniol production. Indeed, we observed that the fraction of geraniol in total monoterpene composition correlated with the ratio between *Ob*GES and t67-*Mc*LIS (*p*-value < 0.05; *t*-test). Together, these findings were encouraging, as they firmly demonstrated that genetic design can be used to control monoterpene production and that the knowledge gained from our initial test set could guide subsequent design.

We next set out to generate a second iteration of designs with target production levels defined by three commercially hopped beers that span a wide range of monoterpene concentrations and perceived hop flavor/aroma intensity. The salient trends observed in the test set informed two guiding principles: (1) to shift overall monoterpene production towards higher levels, designs were composed of strong promoters driving tHMGR, FPPS*, and *Ob*GES and (2) to ensure variation in the ratio of linalool to geraniol, designs encompassed a range of promoter strengths driving expression of t67-*Mc*LIS. (Supplementary Table 11 and Supplementary Fig. 9). We anticipated that applying these design principles towards desired performance characteristics would improve search efficiency. In order to evaluate the extent of improvement, we established a mathematical modeling-based framework to predict the relationship between genetic design and monoterpene production (Online Methods, Supplementary Note 3, and Supplementary Tables 2–5). Using this framework, we predicted that the selected design principles significantly improved our search efficiency (Supplementary Fig. 10). Importantly, we observed a consistent improvement in comparing actual distance-from-target monoterpene levels between the first and second iteration strains (Fig. 4).

**Engineered strains affect consistent hop flavor.** The anticipated commercial value of generating hop flavor molecules through yeast biosynthesis is predicated on three assumptions: (1) because

the conditions inside a fermenter can be precisely controlled, the final concentrations of yeast-biosynthesized monoterpenes in beer are likely to be more consistent compared with those given by conventional hop additions, (2) the biosynthesized monoterpenes linalool and geraniol confer hoppy flavor as perceived through human taste, and (3) the variation in hop flavor molecule concentrations correspond to differences on the order of those discernable by human taste. To test the consistency of yeast-biosynthesized monoterpene levels, replicate fermentations were performed at 8 L scale with a subset of engineered strains. To test the consistency of hop-derived monoterpene levels, Cascade hop preparations originating from five different farms in the Northwestern United States were used to supplement fermentations performed with the parent strain (Fig. 5a,b). We observed little variation in final monoterpene concentrations between replicate samples fermented with engineered strains, whereas fermentations hopped with different preparations yielded significantly greater variation (linalool *p*-value $<1 \times 10^{-5}$, geraniol *p*-value $<1 \times 10^{-3}$; *t*-test) (Fig. 5a,b). This result demonstrates that engineered strains biosynthesize monoterpenes more consistently than can be achieved by randomly selecting Cascade hop preparations from different farms. Next, to test whether yeast-biosynthesized monoterpenes conferred hop flavor, beer was produced in an authentic, pilot-scale brewhouse, following a recipe for a classic American Ale, using three engineered strains and the parent strain (WLP001) as a control (Supplementary Fig. 11 and Supplementary Table 17). A panel of tasters determined that the finished beers exhibited a range of hop flavor/aroma intensity (Fig. 5c). In addition, the apparent difference in hop flavor/aroma intensity between beer fermented with JBEI-16652 and JBEI-16669 was considerable, despite only a ~twofold difference in linalool and geraniol concentrations. Taken together, these results demonstrate that monoterpenes derived from yeast biosynthesis during fermentation give rise to hop flavor/aroma in finished beer and that biosynthesis provides greater consistency than traditional hopping. Finally, in order to compare the intensity of hop flavor conferred by traditional dry hopping with the hop flavor conferred by engineered strains,

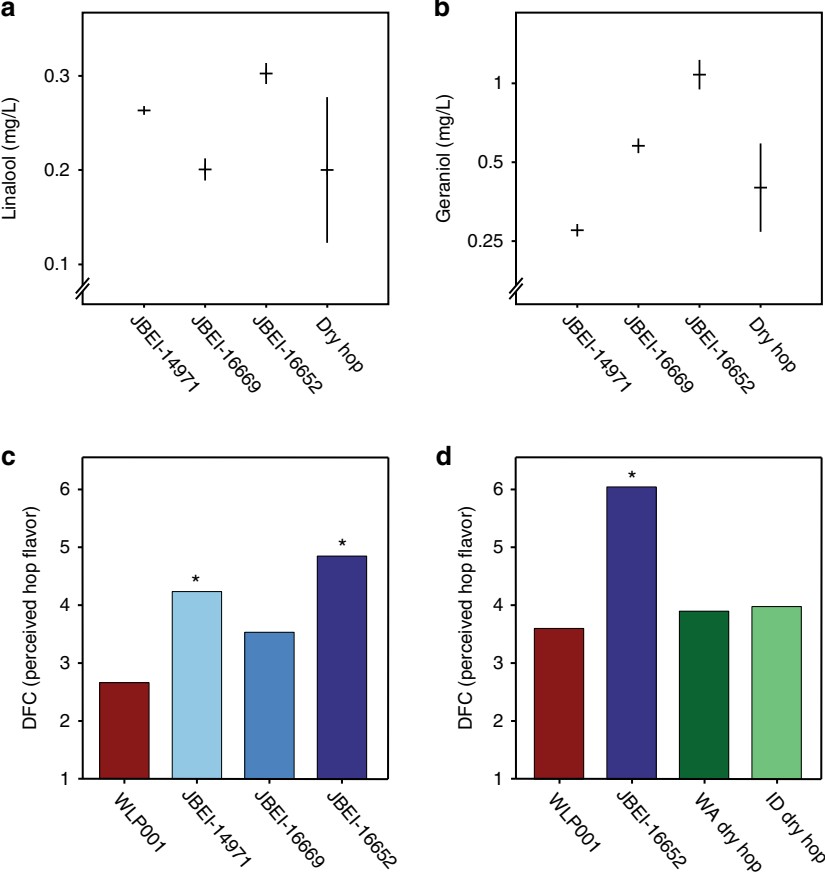

**Fig. 5** Characteristics of pilot-scale beer fermented with engineered strains. **a, b** Variation in linalool (**a**) and geraniol (**b**) concentrations of engineered brewing strain fermentations compared with variation in concentrations generated by traditional dry hopping. For engineered strain samples, horizontal lines correspond to the mean of three biological replicates. For traditional dry hopping, the horizontal line corresponds to the mean of five Cascade hop samples obtained from different farms. Vertical lines correspond to standard deviation. **c** Sensory analysis of the pilot-scale beers fermented with three engineered strains compared to beer fermented with the parental strain. **d** Sensory analysis of pilot-scale beers fermented with engineered strain JBEI-16652 compared to beer fermented with the parental strain, with or without traditional Cascade dry-hopping. Asterisks (**c, d**) indicate statistically significant differences in hop aroma intensity as compared to the control beer ($p$-value < 0.05; Dunnett's test). Difference from control, DFC, was measured on a 9-point scale

fermentations were performed both with the parent strain with dry-hop additions as well as with JBEI-16652 without dry-hop additions (Supplementary Fig. 11 and Supplementary Table 18). Conventionally dry-hopped beers consistently exhibited increased hop flavor/aroma as perceived by a sensory panel; however, these effects were not statistically significant compared to the parental control (Fig. 5d). In contrast, beer produced with JBEI-16652 again exhibited significantly higher hop flavor/aroma than the parental control. Similar monoterpene concentrations were observed between the two batches, demonstrating the consistent performance of the engineered strain.

## Discussion
In this study, we have engineered brewer's yeast for production of flavor molecules ordinarily derived from hops. We developed new methods to overcome the difficulties associated with stable integration of large constructs in industrial strains. Unlike classical microbial metabolic engineering efforts that focus on maximizing the titer of a single molecule, we focused on tuning the expression of key genes in a biosynthetic pathway to simultaneously make precise concentrations of multiple flavor molecules. This application promises to generate hop flavors with more consistency than traditional hop additions, as hop preparations are

notoriously variable in the content of their essential oil and the flavor they impart to beer[23]. It should be noted that blending hop preparations from different sources can be used to reduce variation. However, blending is ultimately limited by practical constraints: In the best case, large craft breweries create one single hop blend per year, which fails to mitigate year-to-year variation. Our strategy is favored over plant or microbial bioprocess extraction because it avoids the use of non-renewable chemicals typical of industrial extraction. While historic consumer trepidation towards genetically engineered foods is of concern for widespread adoption, the general increase in consumer acceptance of such foods when tied to increased sustainability[27] is encouraging.

Previous studies have demonstrated the feasibility of engineering brewer's yeast by incorporation of heterologous genes[13,28,29]; however, the scope and commercial relevance of these efforts have been limited, in part due to methodological difficulties of incorporating an array of large, genetically stable DNA constructs into industrial yeasts. Recent studies have resorted to alternative methods such as breeding hybrid strains[30]. While this has proven to be a powerful approach for generating diverse aroma phenotypes, it is intrinsically limited to enzymes and aromas associated with native yeast metabolism. Here, we developed a complementary methodology that allows for stable

incorporation of plant secondary metabolism genes into industrial brewer's yeast. We provide evidence that incorporating linalool and geraniol biosynthesis confers hop flavor to beer. We note that the full flavor imparted by traditional hopping is likely to rely on a more diverse bouquet of molecules. The methodologies described herein provide a foundation for generating more complex yeast-derived hop flavors, and broaden the possibilities of yeast-biosynthesized flavor molecules to those throughout the plant kingdom.

## Methods

**Cloning**. All strains, expression plasmids, and additional plasmids used for strain construction are listed and described in Supplementary Tables 6–13. The sequence files corresponding to each plasmid can be found in the JBEI Public Registry (https://public-registry.jbei.org/)[31]. Plasmids were propagated in *Escherichia coli* strain DH10B and purified by Miniprep (Qiagen, Germantown, MD, USA). The "pathway" plasmids used to construct the engineered brewing strains were assembled by the standard Golden Gate method using type II restriction enzymes and T7 DNA Ligase (New England Biolabs, Ipswich, MA, USA)[24,32] (for additional detail, see schematized assembly strategy in Supplementary Fig. 3). All other plasmids generated in this study were constructed by Gibson assembly[33] using Gibson assembly master mix (New England Biolabs, Ipswich, MA, USA). Constructs were designed using the DeviceEditor bioCAD software[34], and assembly primers were generated with j5 DNA assembly design automation software[35] using the default settings. PCR amplification was performed using PrimeSTAR GXL DNA polymerase according to the manufacturer's instructions (Takara Bio, Mountain View, CA, USA). Genes coding for full-length linalool and geraniol synthases were ordered either from IDT (San Diego, CA, USA) as G-blocks or from Life Technologies (Carlsbad, CA, USA) as DNA strings. The coding sequences of heterologous genes in all plasmids were validated by Sanger sequencing (Genewiz, South Plainfield, NJ, USA and Quintara, South San Francisco, CA, USA).

**Strain construction**. Yeast lab strains were transformed by the high-efficiency lithium acetate method[36]. Strains were cultivated in yeast extract + peptone + dextrose (YPD) medium unless otherwise noted. To select for transformants containing auxotrophic complementation cassettes, transformed cells were plated on standard dropout medium (Sunrise Science Products, San Diego, CA, USA). To select for transformants containing drug resistance cassettes, cells were recovered in YPD medium for 4 h after transformation, and then plated on YPD medium supplemented with 200 μg/L geneticin (Sigma-Aldrich, St. Louis, MO, USA) or hygromycin B (Sigma-Aldrich, St. Louis, MO, USA). Minor modifications were made to cultivation conditions for brewer's yeast transformations: pre-transformation cultures were grown in YPD medium supplemented with 200 mg/L adenine sulfate at 20 °C in glass test tubes with shaking at 200 rpm. A single colony was used to inoculate an initial 5 mL culture, which was grown overnight to turbidity. This culture was used to inoculate a second 5 mL culture to an $OD_{600}$ (optical density at 600 nm) of 0.01, which was grown for 18 h. The second culture was then used to inoculate 50 mL cultures in 250 mL Erlenmeyer flasks to $OD_{600}$ of 0.05. After ~8 h of growth, strains were transformed by the lithium acetate method[36], cells were recovered in YPD medium for 4 h, plated on YPD supplemented with 200 μg/L geneticin, and then grown for 5–7 days at 20 °C.

DNA used for genomic integration was prepared either by PCR-amplifying plasmid DNA or by digesting a plasmid with restriction enzymes. For construction of the GPP-hyper-producing strain, integration fragments were amplified from the corresponding plasmids by PCR (Supplementary Table 7). For construction of pathway-integrated brewing strains, plasmid DNA was linearized by restriction digestion with *Not*I-HF and *Pst*I-HF (New England Biolabs, Ipswich, MA, USA) (Supplementary Tables 10 and 11).

All integration events were confirmed by diagnostic PCR using GoTaq Green Master Mix (Promega, Madison, WI, USA). For brewer's yeast strains, homozygosity at the integration locus was tested using primers targeted to the 5′ and 3′ junctions of desired allele and the parental allele. The identity of the multi-gene integration was verified with primers targeted to each of the four promoter/gene junctions. The promoter identities corresponding to each strain can be found in Supplementary Tables 12 and 13.

**Screening synthases**. For the linalool and geraniol synthase screening, single colonies were picked from the transformation plate and used to inoculate cultures in 5 mL CSM-Leu (Sunrise) +2% raffinose (Sigma-Aldrich, St. Louis, MO, USA) medium. After 24 h, the precultures were diluted into fresh CSM-Leu + 2% galactose (Sigma-Aldrich, St. Louis, MO, USA) medium to an OD of 0.05 and grown for 72 h with shaking at 200 rpm. An organic overlay was added 24 h after inoculation to capture hydrophobic monoterpenes. Decane was used as the overlay for the cultures expressing LIS and dodecane was used for those expressing GES. The overlay was chosen so as to minimize overlap of retention times between solvent and product for subsequent gas chromatography–mass spectrometry (GC/MS) analysis.

**Microaerobic fermentation**. Strains were streaked on YPD medium and grown for 2 days at 25 °C. Single colonies were used to inoculate initial 2 mL precultures in 24-well plates (Agilent Technologies, Santa Clara, CA, USA), which were grown for 3 days at 20 °C with shaking at 200 rpm. Strains were grown in a base medium composed of 100 g/L malt extract (ME) (Sigma-Aldrich, St. Louis, MO, USA). Each well contained a 5 mm glass bead (Chemglass Life Sciences, Vineland, NJ, USA). The resulting cultures were used to inoculate second 6 mL precultures in fresh 24-well plates to an OD of 0.1, which were then grown for 3 days at 20 °C with shaking at 120 rpm. The resulting cultures were then used to inoculate 25 mL cultures in glass test tubes to an OD of 1.0. These cultures were equipped with a one-way airlock for microaerobic fermentation and grown for 5 days at 20 °C (Supplementary Fig. 5). Test tubes were vortexed for 30 s every 24 h.

**High-performance liquid chromatography**. Maltotriose, maltose, glucose, and ethanol were separated by high-performance liquid chromatography (HPLC) and detected by a refractive index (RI) detector. On day 5, fermentation samples were centrifuged at $18,000 \times g$ for 5 min, filtered using Costar® Spin-X® Centrifuge Tube Filters, 0.22-μm pore, transferred to HPLC tubes, and loaded into an Agilent 1100 HPLC equipped with an Agilent 1200 series auto-sampler, an Aminex HPX-87H ion exchange column (Bio-Rad, Hercules, CA USA), and an Agilent 1200 series RI detector. Metabolites were separated using 4 mM $H_2SO_4$ aqueous solution with a flow rate of 0.6 mL/min at 50 °C. Absolute sample concentrations were calculated using a linear model generated from a standard curve composed of authentic maltotriose, maltose, glucose, and ethanol standards (Sigma-Aldrich, St. Louis, MO, USA) diluted in water over a range of 0.2–20 g/L. All data are provided in Supplementary Table 15.

**Monoterpene quantification**. Monoterpenes were quantified by GC/MS analysis, using an Agilent GC system 6890 series GC/MS with Agilent mass selective detector 5973 network. In all experiments, 1 μL of the sample was injected (split-less), using He as the carrier gas onto a CycloSil-B column (Agilent, 30 m length, 0.25 mm inner diameter (i.d.), 0.25 μm film thickness, cat. no. 112-6632). The carrier gas was held at a constant flow rate of 1.0 mL/min and EMV mode was set to a gain factor of 1.

Sampling, oven temperature schedule, and ion monitoring was optimized for each experiment: for quantifying linalool and geraniol production in terpene synthase screens, the samples were spun down and the organic phase (solvent overlay) was collected, diluted 1:10 in ethyl acetate (Sigma-Aldrich, St. Louis, MO, USA), transferred to a glass GC vial, and injected into the GC column. For samples corresponding to the LIS screen, the oven temperature was held at 50 °C for 12 min, followed by a ramp of 10 °C/min to a temperature of 190 °C and a ramp of 50 °C/min to a final temperature of 250 °C, and then held at 250 °C for 1 min. The solvent delay was set to 20 min, and the MS was set to SIM mode for acquisition, monitoring $m/z$ ions 80, 93, and 121. For samples corresponding to the geraniol synthase screen, the oven temperature was held at 50 °C for 5 min, then ramped at 30 °C/min to a temperature of 135 °C, then ramped at 5 °C/min to a temperature of 145 °C, then ramped at 30 °C/min to a temperature of 250 °C, and held at 250 °C for 1 min. The solvent delay was set to 10.8 min and the MS was set to monitor $m/z$ ions 69, 93, 111, and 123. For quantifying linalool and geraniol in microaerobic fermentations performed with brewer's yeasts, samples were extracted on day 5 using ethyl acetate. Fermentation samples were collected and spun down, 1600 μL of the supernatant was mixed with ethyl acetate at a 4:1 ratio in a 96-well plate, the plate was sealed and vortexed for 2 min, then spun at $3000 \times g$ for 5 min, and 30 μL of the ethyl acetate was transferred into a glass GC vial. The resulting preparation was injected into the GC column. For quantifying linalool and geraniol in various commercial beers, 2 mL of ethyl acetate was added to 8 mL of the beer in glass tubes (Kimble Chase, Rockwood, TN, USA). This was mixed by hand for 2 min and spun at $1000 \times g$ for 10 min. Thirty microliters of the ethyl acetate layer was transferred to glass GC vials, and the resulting preparation was injected into the GC column. For both the microaerobic fermentation experiments and sampling of commercial beers, the oven temperature was held at 50 °C for 5 min, followed by a ramp of 5 °C/min to a temperature of 200 °C and a ramp of 50 °C/min to a final temperature of 250 °C, and then held at 250 °C for 1 min. The solvent delay was set to 5 min and the MS was set to monitor $m/z$ ions 55, 69, 71, 80, 81, 93, 95, 107, 121, 123, and 136.

Peak areas for linalool and geraniol were quantified using MSD Productivity ChemStation software (Agilent Technologies, Santa Clara, CA, USA). Absolute sample concentrations were calculated using a linear model generated from a standard curve composed of authentic linalool and geraniol standards (Sigma-Aldrich, St. Louis, MO, USA). For monoterpene synthase screening experiments, standards were diluted in ethyl acetate over a range of 0.2–50 mg/L. For the microaerobic fermentation experiments and sampling of commercial beers, standards were spiked into a preparation extracted from the parent strain fermentation sample (i.e., a control preparation used to ensure accurate baseline signal) over a range of 0.2–10 mg/L. In calculating actual concentrations, apparent concentrations were scaled based on dilution or concentration in GC injection preparation.

**Proteomics**. Protein abundance data are reported in Supplementary Table 16. Culture (5 mL) was sampled after 2 days, vortexed, and spun at $3000 \times g$ for 5 min. The supernatant was discarded, and the pellet was flash frozen. Plate-based cell pellets were lysed by chloroform-methanol precipitation, described below, while samples in tubes were lysed by re-suspending the pellets in 600 μL of yeast lysis buffer (6 M urea in 500 mM ammonium bicarbonate), followed by bead beating with 500 μL zirconia/silica beads (0.5 mm diameter; BioSpec Products, Bartlesville, OK, USA). Samples in tubes were bead beat for five cycles of 1 min with 30 s on ice in between each cycle. Subsequently, they were spun down in a benchtop centrifuge at a maximum speed for 2 min to pellet cell debris, and the clear lysate was transferred into fresh tubes. Plate-based cell lysis and protein precipitation was achieved by using a chloroform-methanol extraction[37]. The pellets were re-suspended in 60 μL methanol and 100 μL chloroform, and then 50 μL zirconia/silica beads (0.5 mm diameter; BioSpec Products, Bartlesville, OK, USA) were added to each well. The plate was bead beat for five cycles of 1 min with 30 s on ice in between each cycle. The supernatants were transferred into a new plate and 30 μL water was added to each well. The plate was centrifuged for 10 min at a maximum speed to induce the phase separation. The methanol and water layers were removed, and then 60 μL of methanol was added to each well. The plate was centrifuged for another 10 min at a maximum speed and then the chloroform and methanol layers were removed and the protein pellets were dried at room temperature for 30 min prior to re-suspension in 100 mM ammonium bicarbonate with 20% methanol.

The protein concentration of the samples was measured using the DC Protein Assay Kit (Bio-Rad, Hercules, CA, USA) with bovine serum albumin used as a standard. A total of 50 μg protein from each sample was digested with trypsin for targeted proteomic analysis. Protein samples were reduced by adding tris 2-(carboxyethyl)phosphine to a final concentration of 5 mM, followed by incubation at room temperature for 30 min. Iodoacetamide was added to a final concentration of 10 mM to alkylate the protein samples and then incubated for 30 min in the dark at room temperature. Trypsin was added at a ratio of 1:50 trypsin:total protein, and the samples were incubated overnight at 37 °C.

Peptides were analyzed using an Agilent 1290 liquid chromatography system coupled to an Agilent 6460 QQQ mass spectrometer (Agilent Technologies, Santa Clara, CA, USA). The peptide samples (10–20 μg[LC2]) were separated on an Ascentis Express Peptide ES-C18 column (2.7 μm particle size, 160 Å pore size, 5 cm length × 2.1 mm i.d., coupled to a 5 mm × 2.1 mm i.d. guard column with similar particle and pore size; Sigma-Aldrich, St. Louis, MO, USA), with the system operating at a flow rate of 0.400 mL/min and column compartment at 60 °C. Peptides were eluted into the mass spectrometer via a gradient with initial starting condition of 95% Buffer A (0.1% formic acid) and 5% Buffer B (99.9% acetonitrile, 0.1% formic acid). Buffer B was held at 5% for 1.5 min, and then increased to 35% Buffer B over 3.5 min. Buffer B was further increased to 80% over 0.5 min where it was held for 1 min, and then ramped back down to 5% Buffer B over 0.3 min where it was held for 0.2 min to re-equilibrate the column to the initial starting condition. The peptides were ionized by an Agilent Jet Stream ESI source operating in positive-ion mode with the following source parameters: gas temperature = 250 °C, gas flow = 13 L/min, nebulizer pressure = 35 psi, sheath gas temperature = 250 °C, sheath gas flow = 11 L/min, VCap = 3500 V. The data were acquired using Agilent MassHunter, version B.08.00. Resultant data files were processed by using Skyline[38] version 3.6 (MacCoss Lab, University of Washington, Seattle, WA, USA) and peak quantification was refined with mProphet[39] in Skyline.

**Data analysis**. Data analysis was performed using the R statistical programming language[40]. Additional libraries were used for data visualization functionalities[41–45]. For protein and metabolite analysis heatmaps (Fig. 2e and Supplementary Fig. 9), relative levels were reported as follows: promoter strengths were represented as a fraction of their previously reported rank order[24] ranging from $P_{RNR2}$ (0) to $P_{TDH3}$ (1). Feature scaling was used to standardize the range of protein, monoterpene, and sugar abundances. Let $s_i$ equal the $\log_{10}$-transformed abundance value for species $S$ in strain $i$. Normalized values were computed according to Eq. (1) as:

$$s_i^{'} = \frac{s_i - \min(S)}{\max(S) - \min(S)} \tag{1}$$

For sugar analysis, unfermented ME was included in max/min calculations. For fermentable sugars (i.e., maltotriose, maltose, glucose), the scaled values were subtracted from 1 in order to represent proximity to desired sugar consumption profile.

The distance metric of an engineered strain with respect to a given commercial beer was calculated using the Manhattan length as the distance of monoterpene production from beer monoterpene concentrations and the distance in sugar consumption from the parent strain. First, the difference between $\log_{10}$-transformed values of engineered strain monoterpene concentration and target beer monoterpene concentration was calculated for each species, linalool and geraniol. Second, the absolute values of these differences were calculated. Finally, the resulting values, together with the fraction of total sugar remaining after fermentation, were averaged.

**Mathematical modeling**. Three different models were constructed in Python to predict monoterpene production from protein levels (for detailed description and implementation, see Supplementary Data File 1). Files containing data used to generate predictive models are included as Supplementary Data Files 2 and 3. Both the Gaussian regressor and linear models were implemented using Scikit-learn[46]. Additional equations needed to describe the linear model are given in Supplementary Table 3. Equations describing the Michaelis–Menten kinetics model are given in Supplementary Table 4 and a schematic of the model structure is provided in Supplementary Fig. 13. Kinetic parameters were scraped from the literature (Supplementary Table 5) and protein concentrations are given in Supplementary Data File 2. Free parameters were included to convert relative protein counts to absolute protein values. Additionally, a parameter $\beta$ determined the relative ratio between the endogenous FPPS and FPPS*.

Both the linear and Gaussian regressor models were fit using standard methods from the Scikit Learn library. The kinetic model was manually constructed without external libraries. To fit the kinetic model, a differential evolution algorithm was used to perform parameter optimization on a nonlinear cost function. Specifically, the sum of the squared residual error of the model predictions from the first iteration strains was minimized with respect to the previously described parameters. The kinetic coefficients were bounded to vary over an order of magnitude from the values described in the literature. In order to cross-validate the models and minimize overfitting, a leave-one-out methodology was applied to each model. The error residuals from this cross-validation technique are reported in Supplementary Table 2.

Analysis performed for predicting the extent of performance improvement for second iteration strains (Fig. 4 and Supplementary Fig. 10) compared with randomly designed strains is described in Supplementary Note 3.

**Toxicity assay**. $OD_{600}$ measurements were taken in 48-well clear flat bottom plates (Corning Inc., Corning, NY, USA) using a Tecan Infinite F200 PRO reader, with acquisition every 15 min. Analysis was performed using custom python scripts. Growth curves were calculated by averaging six biological replicates; shaded areas represent one standard deviation from the mean. Growth rates were calculated with a sliding window of 5 h, solving for maximum growth rate. Growth rates are presented as the average of six biological replicates; error bars represent 95% confidence intervals (Supplementary Fig. 12).

**Pilot fermentations**. Strains were streaked on YPD medium and grown for 2 days at 25 °C. Single colonies were used to inoculate initial 5 mL cultures in glass test tubes, which were grown for 2 days at 20 °C with shaking at 200 rpm. The resulting cultures were used to inoculate 1 L cultures in 2 L glass Erlenmeyer flasks, which were then grown for 3 days at 20 °C with shaking at 200 rpm. Strains were grown in a base medium composed of 100 g/L ME (Sigma-Aldrich, St. Louis, MO, USA). The resulting cultures were then used to inoculate industrial fermentations in wort produced in a 1.76 hL pilot brewery.

For the first set of fermentations, 35 kg of 2-Row malt was milled and added to 105 L of DI water treated with 79.15 g of brewing salts. Mashing was performed for 30 min at 65 °C, 10 min at 67 °C, and 10 min at 76 °C. The wort was allowed to recirculate for 10 min and was separated by lautering. Sparging occurred for 58 min, giving a final pre-boil volume in the brew kettle of 215 L. The wort was boiled until it reached a final volume of 197 L and a gravity of 11.65 °Plato. Kettle additions included 125 g of Magnum hop pellets, 15.1 g of Yeastex yeast nutrients, and 15 g Protofloc (Murphy and Son, Nottingham, UK). Ingredients were sourced from Brewers Supply Group (Shakopee, MN, USA), except where otherwise noted. After the wort was separated from the hot trub, it was transferred to four 56 L custom fermenters (JVNW, Canby, OR, USA), each filled to 40 L. The beers were fermented at 19 °C until they reached terminal gravity, held for an additional 24 h for vicinal diketone (VDK) removal, and then cold conditioned at 0 °C. The length of fermentation, and in turn the length of cold conditioning, was strain dependent. Samples were taken every 24 h to measure °Plato and pH (see Supplementary Fig. 11). The resulting beer was filtered under pressure and carbonated prior to storage in 7.75 gallon kegs. Samples were collected during the kegging process for Alcolyzer (Anton Paar, Ashland, VA) analysis (see Supplementary Table 17).

For the second set of fermentations, 35 kg of 2-Row malt was milled and added to 105 L of DI water treated with 79 g of brewing salts. Mashing was performed for 30 min at 65 °C, 10 min at 67 °C, and 10 min at 76 °C. The wort was allowed to recirculate for 10 min and was separated by lautering. Sparging occurred for 52 min, giving a final pre-boil volume in the brew kettle of 214 L. The wort was boiled until it reached a final volume of 194 L and a gravity of 11.25 °Plato. Kettle additions included 97.01 g of Galena hop pellets, 15.1 g of Yeastex yeast nutrients, and 15 g Protofloc (Murphy and Son, Nottingham, United Kingdom). Ingredients were sourced from Brewers Supply Group (Shakopee, MN) except where otherwise noted. After the wort was separated from the hot trub, it was transferred to four 56 L custom fermenters (JVNW, Canby, OR), each filled to 40 L. The beers were fermented at 19 °C until they reached terminal gravity, held for an additional 24 h for VDK removal, and then cold conditioned at 0 °C. The length of fermentation, and in turn the length of cold conditioning, was strain dependent. Samples were taken every 24 h to measure °Plato and pH (see Supplementary Fig. 11). After 48 h at 0 °C, 88.5 g Cascade dry hops (either from Washington or from Idaho) were added to two fermenters containing parent strain WLP001. The dry hops were left

on the beer at 1.67 °C for 1 week before filtering. The resulting beer was filtered under pressure and carbonated prior to storage in 7.75 gallon kegs. Samples were collected during the kegging process for Alcolyzer (Anton Paar, Ashland, VA, USA) analysis (see Supplementary Table 18).

**Sensory analysis**. Institutional Review Board approval for human research was obtained from the UC Berkeley Office for Protection of Human Subjects (CPHS protocol number 2017-05-9941). The Committee for Protection of Human Subjects reviewed and approved the application under Category 7 of federal regulations.

Panelists: Sensory analysis of the brewed beer was conducted at Lagunitas Brewing Company (Petaluma, CA, USA). The first panel consisted of 27 employee participants (17 males and 10 females), the second of 13 employee participants (11 males and 2 females), ranging in experience from 2 to 154 tasting sessions attended in calendar year 2017. Ages ranged from mid-20s to 50s. All participants received basic sensory training per Lagunitas standards.

Sensory analysis: Samples of 2 ounces were presented in clear 6 oz brandy glasses (Libbey, Toledo, OH, USA). Each panelist received five glasses, one control and four samples (one blind control and three variables) arranged randomly by balanced block design. Block design and data gathering were accomplished using the EyeQuestion® software (Logic8 BV, The Netherlands). In a single sitting, panelists were asked to rank hop aroma intensity as compared to the control on a 9-point ordinal scale anchored on one end with "No difference" and the other end with "Extreme difference."

Data analysis: Data were analyzed using Dunnett's test in conjunction with one-way analysis of variance using EyeOpenR® (Logic8 BV, The Netherlands). Analysis was performed at the 95% confidence level. The blind control is used as the reference sample to account for any scoring bias that might occur.

**Dry hopping for evaluation of variation between hop preparations**. Parental strain WLP001 was streaked on YPD medium and grown for 2 days at 25 °C. A single colony was used to inoculate an initial 50 mL preculture in a 250 mL glass Erlenmeyer flask, which was grown for 1 day at 20 °C with shaking at 200 rpm. The strain was grown in a base medium composed of 100 g/L ME (Sigma-Aldrich, St. Louis, MO, USA) supplemented with YPD. The resulting culture was used to inoculate a 1 L preculture in a 2 L glass Erlenmeyer flask, which was then grown for 2 days at 20 °C with shaking at 200 rpm. The resulting culture was then used to inoculate four 2 L cultures in 4 L glass Erlenmeyer flasks, which were then grown for 1 day at 20 °C with shaking at 200 rpm. The resulting cultures were then used to inoculate 8 L cultures in 3-gallon glass carboys (Midwest Supplies, Roseville, MN, USA). These cultures were equipped with a one-way airlock for microaerobic fermentation and grown for 6 days at 20 °C. In the meantime, five different Cascade hop samples grown on farms across the Pacific Northwest were obtained from YCH Hops (Yakima, WA, USA). The hop samples were ground using a mortar and pestle and liquid nitrogen. On day 6, samples were taken from the fermentations (as un-hopped controls) and 25 g of hops was added to each fermentation. Hops were left to steep for 3 days, after which samples were collected for GC/MS analysis.

**Batch-to-batch variation**. Strains were streaked on YPD medium and grown for 2 days at 25 °C. Single colonies were used to inoculate initial 5 mL precultures in glass test tubes, which were grown for 2 days at 20 °C with shaking at 200 rpm. Strains were grown in a base medium composed of 100 g/L ME (Sigma-Aldrich, St. Louis, MO, USA). The resulting cultures were used to inoculate 500 mL precultures in 2 L glass Erlenmeyer flasks, which were then grown for 1 day at 20 °C with shaking at 200 rpm. The resulting cultures were then used to inoculate 8 L cultures in 3-gallon glass carboys (Midwest Supplies, Roseville, MN, USA). These cultures were equipped with a one-way airlock for microaerobic fermentation and grown for 12 days at 20 °C. Samples were taken on day 12 for GC/MS analysis.

**Data availability**. The authors declare that all data supporting the findings of this study are available within the paper and its supplementary information files. Sequence data and strains generated in this study have been deposited in the JBEI public registry. See Supplementary Tables 6–13 for construct sequences and strain information. Computer code used in this study can be accessed from Supplementary Data 1.

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

## Acknowledgements

We thank W. DeLoache, M. Lee, B. Cervantes, and J. Dueber for providing yeast genetic toolkit parts; YCH Hops for providing hop samples and Lagunitas Brewing Company for conducting the sensory analysis; V. Chubukov for providing python scripts to process yeast growth assay data. We also thank M. Brown, D. Gillick, and P. Shih for helpful discussions and comments on the manuscript. This work was funded by NSF grant 1330914 and was part of the DOE Joint BioEnergy Institute (http://www.jbei.org), which was supported by the U.S. Department of Energy, Office of Science, Office of Biological and Environmental Research, through contract DE-AC02-05CH11231 between Lawrence Berkeley National Laboratory and the U.S. Department of Energy. This work was also funded by SBIR grant 1722376.

## Author contributions

C.M.D. and R.A.L. conceived the idea, performed experiments, and wrote the manuscript; V.T.V. performed LIS activity screen; Z.C. developed predictive mathematical models; W.L. performed cloning and strain construction; L.J.G.C. and C.J.P. performed targeted proteomics; J.W. performed the pilot fermentations; B.D. performed the sensory analysis; H.V.S., H.G.M., and C.W.B. supervised the work; J.D.K. conceived the idea, supervised the work, and wrote the manuscript.

## Additional information

**Competing interests:** J.D.K. has a financial interest in Amyris, Lygos, Constructive Biology, and Demetrix, none of which will commercialize this technology. J.D.K, C.M.D., and R.A.L. have submitted a patent application covering aspects of this technology. C.M.D. and R.A.L. have a financial interest in Berkeley Brewing Science. The remaining authors declare no competing interests.

