## [Peer Review File · Nature Communications]

PEER REVIEW FILE

Reviewers' Comments:

Reviewer #4 (Remarks to the Author):

The revised manuscript and rebuttals by Denby et al has attempted to address my previous concerns, however there remain several areas of the manuscript wherein I have a continuing disagreement in opinions regarding their methodology and findings.

The authors state in the manuscript that “incorporating linalool and geraniol biosynthesis alone reconstituted hop flavor in beer”, but then make a note regarding “full flavor” not being achieved. They make a longer statement in their rebuttal regarding the nuances between “hop flavor” and “full hop flavor”. However, I still believe that it should not be claimed that it is possible to replace hops in brewing without the sensory data to show that other hop characteristics do not matter. This needs to be done in a situation where the monoterpene levels of the naturally-hopped versus yeast-hopped beers are comparable, rather than that seen in this study where the dry-hopped beer had much lower linalool and geraniol levels than the engineered beers.

Specific comments:

line 58. As far as I am aware, in ref[13] monoterpenes were synthesized in actual wine, not selective media

line 64. LIS was also expressed in wine yeast previously and should be referenced

line 80. ObGES was used in ref[13] and this should be noted

line 181. How rigorous was the sensory set-up given that the dry-hopped beers were not significantly different to the control. Surely the ability of the panel to accurately distinguish dry-hopped from un-hopped beer should be a minimum requirement before further sensory evaluation could take place. Perhaps too little hops were added (the dry hopped mean was always at the very lower bound of monoterpene levels produced by the yeast strains in Fig 3c)?

line 204. Breeding of hybrid strains is generally used to circumvent problems with engineering industrial yeast strains, but is used to ensure that the resultant strains are non-GMO and therefore

available for use in the EU and elsewhere)and to avoid negative consumer perception of GMO products).

Response to referees (NCOMMS-17-20239A)

Reviewer #4 (Remarks to the Author):

The revised manuscript and rebuttals by Denby et al has attempted to address my previous concerns, however there remain several areas of the manuscript wherein I have a continuing disagreement in opinions regarding their methodology and findings.

The authors state in the manuscript that “incorporating linalool and geraniol biosynthesis alone reconstituted hop flavor in beer”, but then make a note regarding “full flavor” not being achieved. They make a longer statement in their rebuttal regarding the nuances between “hop flavor” and “full hop flavor”. However, I still believe that it should not be claimed that it is possible to replace hops in brewing without the sensory data to show that other hop characteristics do not matter. This needs to be done in a situation where the monoterpene levels of the naturally-hopped versus yeast-hopped beers are comparable, rather than that seen in this study where the dry-hopped beer had much lower linalool and geraniol levels than the engineered beers.

We take reviewer 4’s point that using the word reconstitute is misleading. We have changed the language in the manuscript to reflect this point. There is a difference between conferring a flavor characteristic of natural dry hopping and fully reconstituting a flavor from the natural source. We provide evidence that linalool and geraniol are capable of conferring such flavor but we are aware that these two molecules alone do not fully reconstitute a given hop aroma. Indeed, we show that to do so would be to chase a moving target.

Regarding reviewer 4’s final point, we used a standard industrial dry-hopping rate typical of a pale ale in pilot fermentations. The purpose of this experiment is to provide a context to the intensity of hop aroma conferred by this particular engineered strain.

Specific comments:

line 58. As far as I am aware, in ref[13] monoterpenes were synthesized in actual wine, not selective media *Our understanding based on the methods section of this paper is that yeast were propagated in selective media before addition to grape must (in order to maintain expression from a high copy plasmid). This is an impractical strategy for industrial fermentation. We have modified the language in the manuscript to make this distinction absolutely clear.*

line 64. LIS was also expressed in wine yeast previously and should be referenced *LIS from Clarkia brewerii was previously shown to be active in wine yeast. However, we observed no detectable linalool production from a lab strain expressing this protein from a high copy plasmid (native codon sequence and yeast codon optimized) in a genotype engineered for maximal monoterpene production, and with a GC/MS detection threshold below previously reported concentrations. We chose not to belabor this conflict in experimental results.*

line 80. ObGES was used in ref[13] and this should be noted *Several other studies have used versions of geraniol synthase from basil. We have added a note to supplementary figure 1 clarifying that point.*

line 181. How rigorous was the sensory set-up given that the dry-hopped beers were not significantly different to the control. Surely the ability of the panel to accurately distinguish dry-hopped from un-hopped beer should be a minimum requirement before further sensory evaluation could take place. Perhaps too little hops were added (the dry hopped mean was always at the very lower bound of monoterpene levels produced by the yeast strains in Fig 3c)?

The point of this experiment was to provide context for hop intensity conferred by the engineered strain, as per discussion with the editorial staff at Nature Communications. In conducting this experiment, we chose to use a

standard pale ale hopping regime. We were surprised at how little perceived hop flavor was conferred by dry hop additions. However, given the small effect sizes of dry hopping, it is unsurprising that these results were not statistically significant; therefore, these results should not call into question the rigor of our panel. Regardless of statistical significance, these data still provide the intended context for hop aroma intensity conferred by engineered strains.

line 204. Breeding of hybrid strains is generally used to circumvent problems with engineering industrial yeast strains, but is used to ensure that the resultant strains are non-GMO and therefore available for use in the EU and elsewhere)and to avoid negative consumer perception of GMO products).

We are aware that breeding can be used to generate non-GMO organisms. However, the flavors that can be produced using this approach are still intrinsically limited to those arising from yeast metabolism.